# Effect of Tryptophan Supplementation Levels on the Cecal Microbial Composition, Growth Performance, Immune Function and Antioxidant Capacity in Broilers

**DOI:** 10.3390/metabo15110736

**Published:** 2025-11-11

**Authors:** Xuelan Liu, Chunyan Fu, Qingtao Gao, Heng Zhang, Tianhong Shi, Guiming Li, Yunchao Wang, Yan Shang

**Affiliations:** 1Poultry Institute, Shandong Academy of Agricultural Sciences, Jinan 250100, China; jqsliuxl@163.com (X.L.); fuchunyan1004@126.com (C.F.); qingtao_gao@163.com (Q.G.); 15253179001@163.com (H.Z.); shith2004@163.com (T.S.); truemonk@126.com (G.L.); 2Jinan Key Laboratory of Poultry Germplasm Resources Innovation and Healthy Breeding, Jinan 250100, China; 3Shandong Provincial Key Laboratory of Livestock and Poultry Breeding, Jinan 250100, China; 4Shandong Lanhai Ecological Agriculture Co., Ltd., Dongying 257100, China; wyclanhai@163.com

**Keywords:** broiler, cecal microbiota, metabolism, metagenomic sequencing, growth performance, tryptophan

## Abstract

Background: Tryptophan (Trp) is a limiting amino acid in poultry nutrition. Dietary supplementation of Trp not only enhances production performance, but also supports intestinal barrier integrity, alleviates stress, and boosts immunity, in which the derivatives from gut Trp-metabolizing commensal microbes play crucial roles. However, research on how excessive Trp affects poultry growth, metabolism, and gut microbiota composition remains limited. Methods: In this study, we investigated the effects of varying Trp levels (0.23%, 0.29%, 0.35%) on broiler production performance, immune function, and antioxidant levels through controlled feeding trials. These host responses were further correlated with cecal microbiota metagenomic sequencing data using multivariate analysis. Results: Compared with the basal 0.23% Trp level, a 0.35% of Trp addition significantly impaired broiler body weight gain and feed intake, and 0.29% Trp could increase thymus index and serum superoxide dismutase (SOD) level without affecting the growth performance; neither of these two levels affected the feed conversion rate. The cecal microbial metagenomic data further revealed that Trp supplementation reduced the abundance of harmful bacteria, while increasing the abundance of beneficial bacteria and Trp-metabolizing microorganisms. Correlation analysis showed that Trp supplementation was negatively correlated with body weight (BW) but positively correlated with thymus index and SOD level, with similar trends observed between the abundance of specific Trp-utilizing microorganisms and these indicators. Functional analysis revealed an increase in the abundance of KEGG orthology (KO) related to Trp metabolism from the aforementioned microbes. Conclusions: An appropriate addition of Trp (0.29%) can enhance certain metabolic levels without affecting production performance, which might be achieved through relevant metabolic pathways of intestinal microorganisms.

## 1. Introduction

Tryptophan, an essential amino acid for animals, plays a critical role in the growth and development of poultry. It must be supplied through the diet, as deficiency in practical production leads to issues such as reduced growth rates, poor feather quality, delayed sexual maturity, and decreased egg production [1,2]. As the lowest-concentration amino acid in organisms, Trp readily becomes a rate-limiting substrate in protein synthesis [3]. Furthermore, dietary Trp supplementation alleviates stress responses in poultry [4,5,6], improves intestinal barrier integrity, and enhances immune system activity alongside production performance [2,7,8].

Trp is metabolized in the host primarily through the serotonin (5-hydroxytryptamine) pathway and the kynurenine pathway. Serotonin is a crucial neurotransmitter, and its deficiency in brain tissues leads to abnormal behaviors such as feather pecking, resulting in production losses [9,10,11,12], while the kynurenine pathway metabolizes more than 95% of total Trp [13]. In addition, microorganisms in the gut also utilize part of the dietary Trp to produce a variety of metabolites, including serotonin, indole, and their derivatives [14,15,16,17,18,19,20], thus regulating the availability of the Trp pool in the body and converting Trp to other products to regulate the Trp/kynurenine ratio in the blood [12,21]. Symbiotic microorganisms and the host gut form relative homeostasis in a series of interactions in which metabolic derivatives such as indoles play an important role as signaling molecules [12,19]. In terms of the maintenance of intestinal barrier integrity and immunomodulation, Trp metabolites strengthen the epithelial barrier by binding to the aryl hydrocarbon receptor, a very versatile nuclear receptor in the body, and thus upregulate the expression of related genes such as tight junction proteins and adhesion junction proteins [22,23,24]. At the same time, indole and its derivatives are also involved in the growth and differentiation of immune cells and the expression of cytokines in both non-specific and specific immunity, thus regulating intestinal immunity and suppressing inflammatory responses [25,26]. In summary, Trp has a complex interaction network with host and gut microbes, and changes in Trp levels may cause metabolic perturbations of the whole organism, as well as have an important impact on the structure of the intestinal commensal microbiota.

However, the effect of Trp supplementation on the feed intake and production performance of broilers is inconsistent as concluded by the literature [27,28,29,30]. Moreover, these articles do not pay much attention to other indicators apart from production performance. Additionally, there are more feeding experiments conducted after the 21-day age, while experiments from hatching to 21 days are relatively fewer [2,3,29]. Furthermore, the mechanism by which Trp supplementation affects the growth and metabolism of broilers, as well as the role of intestinal microorganisms in this process, has been insufficiently explored. Therefore, this study employs gradient Trp supplementation combined with metagenomic sequencing to analyze its effects on cecal microbiota and to investigate correlations between microbial community structure and growth/metabolic parameters in order to explore the reasonable dosage of Trp.

## 2. Materials and Methods

### 2.1. Feeding, Management and Treatment

A one-factor experimental design was used and 300 day-old male Abbott-Alaska broilers with similar body weights were randomly divided into 3 groups, with 5 replicates in each group and 20 birds in each replicate. The control (CT) group was fed a basal diet according to the Nutrient Requirement of Broilers in the Feeding Standard of Chicken (NY/T 33-2004) with the Trp level of 0.21% (Table 1, adjusted to 0.23% after actual measurement due to the high content of Trp in the raw materials). The Trp gradient was set based on data from recent studies, which had an upper limit of 0.25–0.27% [31,32,33]. Hence, a low Trp level was established at 0.23% + 0.06% (Trp_low), while a high level was established at 0.23% + 0.12% (Trp_high) to avoid excessive feed costs and imbalanced Trp/Lys ratios (>19%), as reported by Linh (2021) and Baker (2002) [31,34]. All birds were raised in cages with the dimensions of 1.4 m (L) × 0.8 m (W) × 0.4 m (H) per unit (20 birds per cage, group density = 17.8), each equipped with a 1.4 m-long feed trough and two drinkers. The cages were arranged in two tiers: the lower tier was 0.5 m above ground, with a 0.1 m vertical spacing between tiers. Manure was removed daily via a conveyor belt beneath the cages. Feeding occurred twice daily (morning and evening), with temperature initially maintained at around 34 °C in the first week, followed by a weekly reduction of approximately 4 °C thereafter. Continuous lighting (24 h/day) was provided during the first week, with photoperiod reduced progressively by 1–2 h/week thereafter.

### 2.2. Sample Collection

Before sampling, the total weight of each replicate group was measured and the average weight was calculated 12 h after stopping feeding. Given the rapid hyperplasia phase of immune organs (particularly thymus and bursa) prior to 21 days of age, combined with the acute sensitivity of early-stage broilers to external interventions, cecal samples were collected on day 21, with 15 animals selected per group (5 replicates; 3 animals with weights close to the replicate’s average were randomly selected and pooled as one sample) for metagenomic sequencing. As for the measurement of growth performance and biochemical indicators, one representative chicken from these three was selected for subsequent assays.

Sufficient feed ration was prepared before the experiment began, and the remaining and the initial diet weight were recorded to calculate average daily feed intake (ADFI) [(diet weight at day 1 − diet weight at day 21)/21]. The hatch weight and BW at day 21 of each sample was measured to determine body weight gain (BWG) [BW at day 21 − BW at day 1] and average daily gain (ADG) (BWG/21). The feed conversion rate (FCR) was calculated as average daily feed intake divided by average daily gain [ADFI/ADG].

All metal instruments were subjected to high-pressure sterilization before sampling. The chickens were sacrificed by cervical dislocation. The abdominal cavity was opened, and the parenchymal organs were collected from the outside to the inside, followed by the collection of intestinal organs. The cecum was exposed, and this was cut and squeezed to collect the contents into a cryotube, which was then stored in liquid nitrogen.

Organ indices (thymus, liver, bursa of Fabricius, and spleen) were derived by dividing the fresh organ weight by live body weight. Prior to slaughter, all chickens underwent wing venipuncture. After standing at room temperature for 30 min, blood samples were centrifuged at 3000 rpm for 10 min to obtain serum. The activities of GSH-Px, SOD, T-AOC, and MDA, as well as antibody (IgA, IgG, IgM) titers, were measured using commercial kits (Jiyinmei Biotech, Wuhan, China) following manufacturer’s protocols.

### 2.3. Data Processing and Analyses

Growth and biochemical data were first sorted and subjected to the Grubbs’ critical value test, then analyzed by one-way ANOVA in SPSS v.19. Homogeneity of variance tests were used to assess the variance across different groups, followed by Duncan’s test to analyze the sources of inter-group differences. Metagenomic libraries were sequenced on an Illumina NovaSeq 6000 system (Illumina, Inc., San Diego, CA, USA) (150 bp paired-end). After adapter trimming and quality control (Trimmomatic; ILLUMINACLIP: adapters_path: 2:30:10 SLIDINGWINDOW: 4:20 MINLEN: 50), host DNA was depleted via alignment to the Gallus gallus GRCg6a reference genome (Bowtie2; sensitivity:—very-sensitive).

In order to study the species composition and diversity information of the samples, we used Kraken2 (parameters—confidence 0.2) to annotate and classify all the valid sequences of all the samples. Based on the absolute abundance and annotation information from Bracken, we calculated the proportion of the number of sequences in each sample in the total number of sequences across a total of seven categories. The OTU data were subjected to cloudtutu.com.cn for α-diversity analyses (Shannon, Simpson, Chao1 and ACE index), the Wilcoxon rank-sum test was used for comparing two independent groups, and the ANOVA overall test was employed for assessing global differences across three groups. β-diversity was calculated based on the Bray–Curtis distance matrix to determine the differences between samples, and is presented in a reduced-dimensional format through Principle Coordinates Analysis (PCoA). Using the HUMAnN 2.0 software, the sequences after quality control and de-hosting were compared with the protein database UniRef90. Reads that failed the comparison were filtered out (default comparison parameters: translated query coverage threshold = 90.0, prescreen threshold = 0.01, evalue threshold = 1.0, translated subject coverage threshold = 50.0). The relative abundance of each protein in UniRef90 was calculated to perform the LEfSe analysis, redundancy analysis (RDA) (https://bioincloud.tech/task-meta, accessed 23 April 2025), and KEGG pathway analysis and coloring.

## 3. Results

### 3.1. Effects of Dietary Trp Levels on Growth Performance, Organ Development and Serum Biochemical Indicators in Broilers

Compared with the control group and the Trp_low group, 0.35% of Trp significantly reduced the BW, BWG, ADFI, and ADG in 21-day-old broilers (*p* < 0.05), while 0.29% of Trp had no negative effects on BW, BWG, ADFI, and ADG. However, both additional levels had no significant effect on FCR according to Table 2. The 0.29% and 0.35% of Trp dose-dependently and significantly increased the thymus index in 21-day-old broilers; the 0.35% Trp addition significantly reduced the liver index and the bursa of Fabricius index, whereas neither dosage significantly altered the spleen index (Table 3). In addition, both 0.29% and 0.35% dietary Trp supplementation showed no significant effects on serum glutathione peroxidase activity and total antioxidant capacity as well as immune globulin titers, but slightly reduced malondialdehyde (a lipid peroxidation product) levels while significantly enhancing SOD activity in serum (Table 4). In general, 0.29% addition of Trp could enhance the development of immune organs (thymus, bursa of Fabricius) without affecting production performance while also benefiting serum SOD levels.

### 3.2. Effects of Dietary Trp Levels on Cecal Microbiota Composition and Diversity Analysis

Analysis of species distribution across three metagenomic sample groups revealed that the predominant microbial phyla (relative abundance > 5%) were Firmicutes, Bacteroidetes, Verrucomicrobia, and Proteobacteria, whereas variations existed between groups (Figure 1A and Appendix A). Although there was no significant difference compared to the CT group, the Trp_high group exhibited decreased abundances of Firmicutes (13.3%) and Proteobacteria (63.4%) but increased abundances of Bacteroidetes (36.2%) and Verrucomicrobia (98.0%). The Trp_low group showed reduced abundance of Proteobacteria (58.5%) and elevated Verrucomicrobia (216%) (Appendix A). At the genus level, genera of Firmicutes accounted for 17 out of 20 genera with the highest relative abundance, with Lactobacillus accounting for over 33% abundance. Increased relative abundances were observed in *Alistipes* (38.6% in Trp_high), *Akkermansia* (98.0% in Trp_high and 216% in Trp_low), and *Merdimonas* (108% in Trp_low) (Figure 1B and Appendix A), which contributed predominantly (>90%) to the elevated abundances of their respective phyla. In contrast, decreased relative abundances occurred in Escherichia (65.0% in Trp_high and 59.2% in Trp_low), thus directly driving the reduction in Proteobacteria abundance (Appendix A).

The Trp_high group showed significant clustering distinct from other groups, while no notable difference was observed between the CT group and the Trp_low group (Figure 2B). These findings are mutually corroborated by Venn diagram analysis (Figure 3C), which revealed that there were only 10 species unique to the Trp_high group, with merely 16 and 9 species sharing with the control or Trp_low groups, respectively—significantly lower than corresponding data from other groups.

Based on the Chao1 index (*p* = 0.06), ACE index (*p* = 0.06), and Shannon index (*p* = 0.45), dietary supplementation of Trp at 0.35% reduced the community richness and diversity (55.4% in Trp_high and 3.2% in Trp_low) of cecal microbiota in broilers (Figure 3A), which corroborates with the results in Figure 2. On the contrary, the Simpson index (*p* = 0.29) suggested that high-Trp diets reduce the proportion of dominant species (e.g., declined relative abundance in Firmicutes and Proteobacteria according to Appendix A) and improve evenness in cecal microbiota (Figure 3A).

β-diversity analysis indicated differences in community distribution between the Trp_high group and the other two groups, and the Trp_low group shares a major similarity with the CT group except for sample Trp13, suggesting that the high Trp diet restructured the cecal microbial community of broiler chickens (Figure 3B). For biomarkers, the control group exhibited higher abundances of Gemmiger, Clostridium, and Flavonifractor genera and the Trp_low group showed elevated abundances of Christensenella and Provencibacterium genera (Figure 3D, LDA score > 2); the Trp_high group featured only Enterococcus faecalis as a characteristic microorganism with relatively higher LDA score (3.9) (Figure 3E), which might indicate its relatively important roles among the communities.

### 3.3. Correlation Analysis with Growth and Metabolism Indicators

Based on prior laboratory measurements of Trp levels’ effects on broilers, we selected BW, thymus index, spleen index, IgA titers, and SOD activity as environmental variables for RDA to explain its correlation with the distribution of the microbial communities as well as the distribution of Trp groups. Overall, the BW generally exhibited a negative correlation with other environmental variables. Analyses of microbial distribution at the phylum and genus levels revealed no significant correlation between the environmental variables and community structure, with only specific microorganisms such as Bacteroidetes (*Alistipes*), Verrucomicrobia (*Akkermansia*) and *Anaerotignum*, *Anaerotruncus*, and *Enterococcus* showing slightly positive correlations with the SOD index and thymus index (Figure 4A,C). At the phylum level, no significant relationships were observed between three groups and the environmental variables. However, at the genus level, the Trp_high group data exhibited pronounced clustering, demonstrating the negative correlation between high Trp levels and BW, as well as a positive correlation with the thymus index and SOD activity (Figure 4B,D).

The correlation heatmap analysis reveals more obvious interrelationships between environmental variables and certain microbes. At the phylum level, most of the phyla showed positive correlations with BW and negative correlations with other variables, but phyla with relatively high abundance (such as Bacteroidetes, Verrucomicrobia, and Actinobacteria) were all negatively related to BW (Figure 5A). At the genus level, Firmicutes accounted for the majority of the top 29 most abundant genera, and though 22 out of 29 genera positively correlated with BW, most of them came from Firmicutes (19 genera) and Proteobacteria (2 genera). However, the genera of *Akkermansia* (Verrucomicrobia), *Alistipes* (Bacteroidetes), and *Enterococcus* (Firmicutes) were negatively correlated with BW and were positively related to two to three kinds of other variables including SOD, which account for more than 40% of the relative abundance in Trp_low (40.2%) and Trp_high (48.8%) group (Figure 5B and Appendix A).

### 3.4. Functional Analyses of Different Trp Groups

The Kyoto Encyclopedia of Genes and Genomes (KEGG) database was chosen for functional analyses. *Metabolism* accounted for over 70% of the relative abundance in all the KEGG pathways, and the Trp_low and Trp_high group showed significantly higher relative abundance in the two major pathways of Metabolism and Organismal Systems compared to the control group (Appendix A). Forty-five level 2 pathways were identified, among which some pathways belong to the Metabolism, Organismal Systems and Human Diseases showed significant differences in different groups (Appendix A).

Although annotation and analysis of KOs revealed that additional Trp supplementation did not cause significant differences in the relative abundance of microbial KOs in the cecum and reduced the richness (Figure 6A,B), further LEfSe analysis identified that the Trp_high group contained the highest number of significantly different KO terms among three groups (Figure 6C), primarily contributed by *A. muciniphila* and *L. reuteri* (Figure 6D,E), and functional differences in the control group were dominated by *Escherichia coli* and other *lactobacilli* (*Lactobacillus johnsonii*, *L. crispatus*, *L. salivarius*).

In order to explore the relationship between the cecal microorganism function and Trp metabolism, we further statistically analyzed the expression of all the relevant genes within the Trp metabolic pathway among all the samples (Appendix A). A total of 11 key KOs were determined through comparison with the database (Appendix A). Differential expression of related KOs was detected in the pathways where Trp is metabolized into indole, serotonin, kynurenine, tryptamine and their derivatives, such as Trpase in the indole pathway (K01667), amidase in the tryptamine metabolism pathway (K01426), acetaldehyde dehydrogenase in the serotonin metabolism pathway (K00128), catalase in the kynurenine metabolism pathway (K03781), and dehydrogenase (K00658, K00382) in the pathway that generates acetyl-CoA from kynurenine. Among K01667, K03781, K00658, and K00382, the sequences derived from *Akkermansia muciniphila*, *Enterococcus faecalis*, and *Lactobacillus reuteri* accounted for a high proportion of the total detected sequences for the corresponding KOs (Appendix A). More importantly, the abundance of these KOs also significantly increased with the rise in Trp levels, suggesting their important role in the microbial metabolism of Trp.

Beyond Trp metabolism, among the remaining 45 secondary metabolic pathways, numerous subordinate tertiary pathways exist, in which a significant number of relevant KOs were identified from our samples (Appendix A). We thus identified the representative KOs (biomarkers) within the Trp_low and Trp_high groups of these pathways (Appendix A). Most of these KOs come from Metabolism and the rest come from Genetic Information Processing and Cellular Processes. Importantly, these KOs are basically all derived from Akkermansia muciniphila, Enterococcus faecalis, and Lactobacillus reuteri (other species origin with very low abundance were not listed), which is consistent with the source of the key KOs in the Trp metabolic pathway mentioned above (Appendix A). Particularly, enzymes related to nitrogen metabolism such as glutaminase (K01425), the cytochrome enzyme system related to energy metabolism (K00425), and Trpyl-tRNA synthetase (K01867) and DNA primase (K02316) which are related to cell proliferation, are all included in the above-mentioned KOs and their relative abundance also increases with the increase in Trp dosage.

## 4. Discussion

Trp was proven to be able to improve production performance, enhance stress resistance, and regulate intestinal microbiota to bolster immune function. Primarily, in this study, the performance data showed that 0.35% of Trp significantly impairs broiler BW, ADFI, and ADG at 21 days after hatching, and 0.29% of Trp addition retained all these growth performances. This aligns with existing research demonstrating that serotonin—a metabolite synthesized after Trp crosses the blood–brain barrier—suppresses food consumption via negative feedback regulation [35,36]. Xie et al. demonstrated that additional tryptophan significantly decreased feed intake on days 21–35 using Arbor Acres chicks, while diets with higher Trp (0.25%) showed no significant differences in ADFI and ADG compared with lower Trp (0.149–0.162%) addition ones [37], which resembled our results. However, a study conducted on 7 to 21 day broilers (the breed was not mentioned) showed that 0.3% and 0.5% of Trp could increase the BWG and FCR compared to Trp 0.2% [28]. Variations in chicken genetics, dietary formulations, experimental protocols, and rearing conditions, etc., may critically affect outcomes; therefore, prior studies should be interpreted with caution.

Trp supplementation significantly and dose-dependently increased the thymus index, but 0.35% of Trp significantly reduced the liver index as well as the bursa of Fabricius index. Meanwhile, 0.29% and 0.35% of Trp supplementation lifted the antioxidative levels through increased SOD activity and slightly reduced the malondialdehyde concentration, which had been proved in heat stress experiments [5]. Combined with the growth data, supplementing the basal diet with 0.29% of Trp may be appropriate. Additionally, although the thymus and bursa of Fabricius complete most of their structural development within the first three weeks, the potential impact of early intervention on the later functions of immune organs needs to be further verified with data from the 42-day-old stage. Subsequent experiments should extend the sampling period to improve the data.

In this study, dietary supplementation with Trp (0.29%, 0.35%) reduced the abundance of Firmicutes and Proteobacteria (represented by *Escherichia*) in the cecum, while increasing the content of Verrucomicrobiota (*Akkermansia*, especially at 0.29%) and Bacteroidetes (*Alistipes*). Collectively, these shifts indicate a reduction in undesirable bacteria and an increase in beneficial bacteria [38]. Alpha diversity analysis also revealed that high Trp levels reduced the proportion of dominant species while promoting greater species evenness within the community. Consequently, Trp supplementation led to an elevated relative abundance of Trp-metabolizing microbes (e.g., *Alistipes*, *Akkermansia*, and *Enterococcus*) and the formation of distinct community structures [39,40,41,42]. These shifts suggest Trp’s selective pressure on gut microbial metabolic functions, acting as an ecological filter for specific functional taxa [43].

Overall, BW showed a negative correlation with three environmental variables except for IgA, which may find some explanation in the relationship between immune development and self-growth [44,45]. The CT group and Trp_low group showed no significant correlation with several environmental variables, whereas the Trp_high group exhibited a negative correlation with BW, suggesting that excessive Trp may inhibit broiler growth; similar effects have also been observed in studies of other animals [35,36,38,46]. In addition, the Trp_high group and part of the Trp_low group (Figure 4D) were positively related to SOD and thymus indexes. Together, these indicate a correlation between Trp levels and the growth and metabolism of chicken, and this will be discussed in detail in the following sections. And for actual farming practice, a dosage of 0.29% for Trp might be a relatively moderate amount.

Considering the consistency in RDA and correlation analyses at the phylum and genus levels, *Akkermansia* was positively correlated with SOD activity and thymus index, as were some other genera like *Enterococcus*, *Alistipes*, and *Anaerotruncus*. Li et al. also reported that *Akkermansia* was positively correlated with SOD and colon length in mice [47]. *A. muciniphila* is a crucial gut commensal bacterium that plays a significant role in maintaining intestinal barrier integrity [48]. This bacterium has also been demonstrated to participate in physiological activities such as inflammation suppression, immune modulation, and obesity mitigation [49,50,51]. Moreover, researchers have demonstrated a negative correlation between *A. muciniphila* and broiler BW [52], which aligned with the RDA findings in the present study. In addition, experiments conducted on the other above-mentioned species such as *Enterococcus faecalis* also showed beneficial effects on broiler growth performance and immune response [53].

Through the above discussion, Trp exerts selective pressure on the cecal microbial community, thereby increasing the abundance of certain specific microorganisms. Moreover, the level of Trp is also associated with some growth and metabolic indicators of broilers. Therefore, in order to identify the intrinsic relationship among them, we conducted an analysis of the relevant pathways and key KOs in KEGG. Based on the results, the increase in Trp levels leads to enhanced gene expression in certain microorganisms such as *Akkermansia muciniphila* and *Enterococcus faecalis*, including cell proliferation and metabolism. This is manifested macroscopically as an increase in the relative abundance of these microorganisms at various taxonomic levels. Concurrently, the genes related to Trp metabolism in these microorganisms also show increased expression. Previous studies have shown that Trp metabolites such as indole and its derivatives could promote immune organ development and have antioxidative effects, such as increasing the levels of biochemical indicators (SOD and GSH-Px) [54,55], which could partially account for our results.

However, there were limitations in our study. The expression levels of genes related to chicken growth and metabolism, the levels of immune-related cytokines, and the concentration of Trp metabolites were not measured. Therefore, the above inferences lack intermediate evidence, and the correlation among BW, related gene expression in broilers, and Trp metabolism could not be clarified. Meanwhile, the Trp metabolites promote thymus development through AhR, and humoral immunity (mediated by B cells in the bursa) requires T lymphocytes from the thymus for assistance [25]. In this experiment, there was no significant response in humoral immunity while thymus development was enhanced. Due to the lack of measurement of related cytokines such as IL-22, it could only be inferred that this was influenced by the short sampling time and the asynchronous development of the thymus and bursa [56].

## 5. Conclusions

Compared with the basal Trp level, the addition of 0.29% Trp retained the production performance on BW, ADFI, ADG, and FCR of broilers while improving the development of the thymus to a certain extent, increasing the serum SOD level and reducing the MDA level in the early 3 weeks. Conversely, 0.35% of Trp significantly impaired BW, ADFI, ADG, liver index, and bursa of Fabricius index, but increased the thymus index and SOD level. The addition of Trp led to an increase in the abundance of specific Trp-metabolizing microorganisms and the expression of Trp-metabolizing genes in the cecum, and the metabolites produced by these microbes may enhance the development of immune organs such as the thymus and increase the body’s antioxidant levels by activating related receptors; these speculations require a longer experimental period and more complete sample collection to be further refined. In conclusion, 0.29% of Trp in broiler diets maintained the growth performance and enhanced certain metabolic indexes through cecal Trp-utilizing microbes and certain metabolic pathways.

## Figures and Tables

**Figure 1 metabolites-15-00736-f001:**
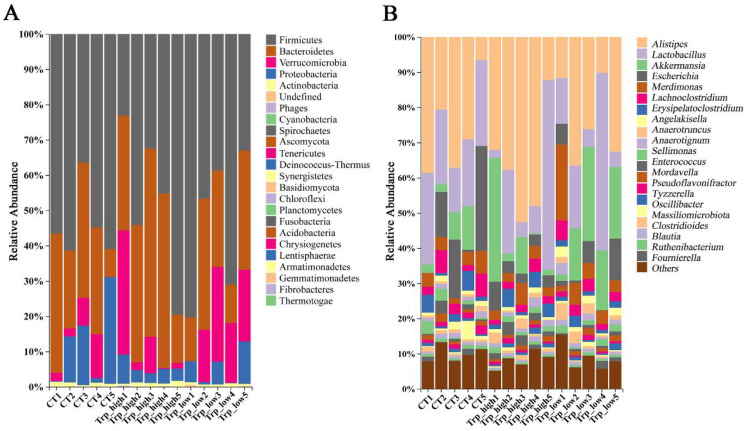
Relative distribution of microbial communities in broiler ceca across different taxonomic levels. (**A**) Relative abundance of species at the phylum level. (**B**) Relative abundance of species at the genus level.

**Figure 2 metabolites-15-00736-f002:**
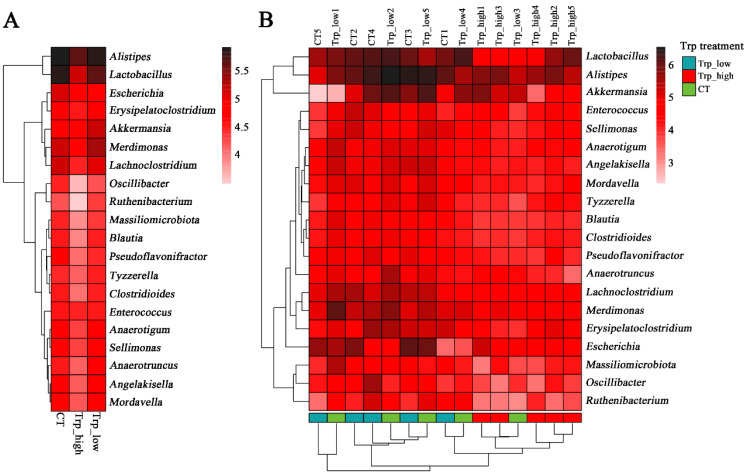
Heatmap analyses of genus-level taxonomic composition of broiler cecal microbiota under different dietary Trp levels. (**A**) Group mean heatmap of top 20 genera. (**B**) Clustering heatmap of top 20 genera; the left panel displays the similarity clustering of species distribution across samples and the bottom is the sample clustering tree.

**Figure 3 metabolites-15-00736-f003:**
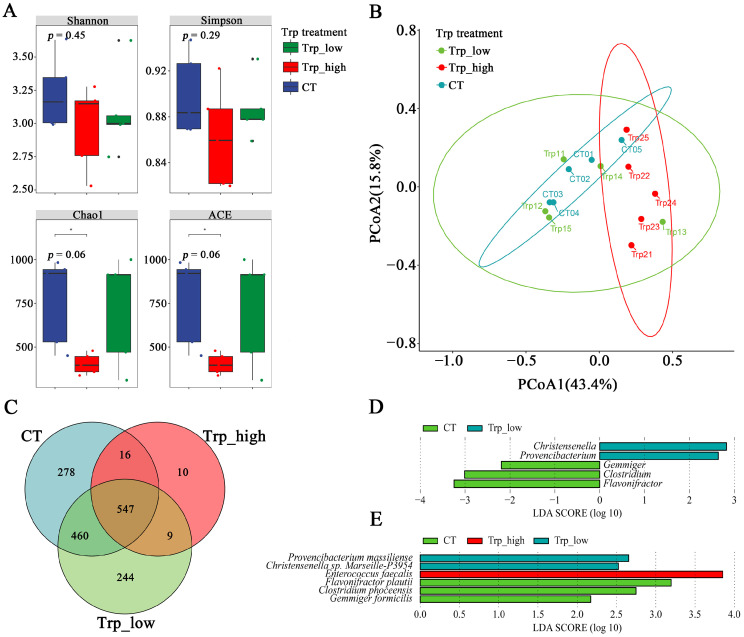
Diversity and significance analysis of different dietary Trp groups. (**A**) α-diversity analyses with Shannon, Simpson, Chao1, and ACE index; the Wilcoxon rank-sum test was used for comparing two independent groups and the ANOVA overall test was employed for assessing global differences across three groups. (**B**) β-diversity analysis using PCoA based on the Bray–Curtis distance matrix. (**C**) Venn diagram analysis of three Trp groups. (**D**) LEfSe analysis of three Trp groups at the genus level. (**E**) LEfSe analysis of three Trp groups at the species level.

**Figure 4 metabolites-15-00736-f004:**
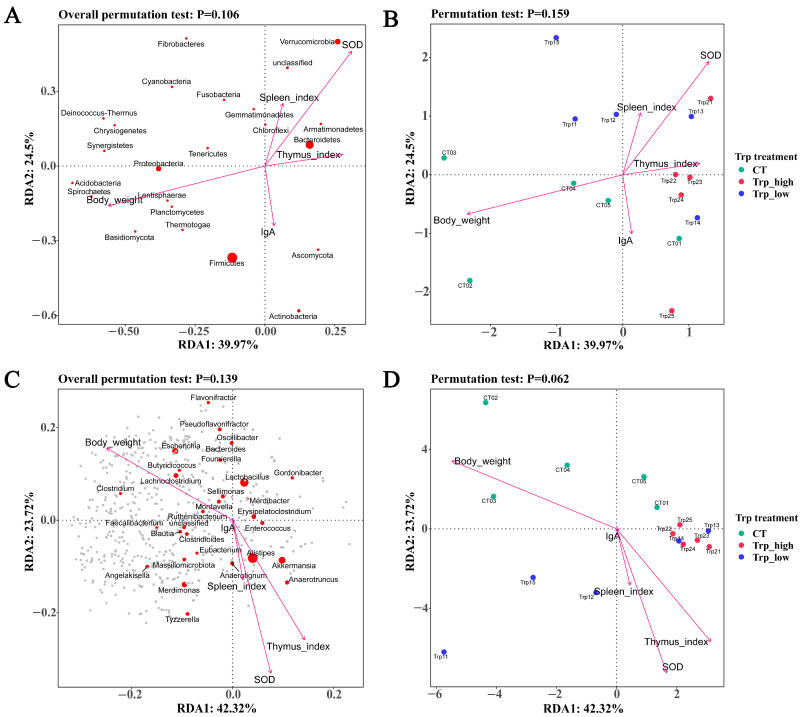
RDA diagrams of relation between environmental variables and microbial communities or Trp groups. (**A**,**C**) RDA diagrams of environmental variables and microbial communities at phylum level; each red dot represents one phylum and the size indicates abundance, while gray dots represent species with lower abundance. (**B**,**D**) RDA diagrams of environmental variables and Trp groups.

**Figure 5 metabolites-15-00736-f005:**
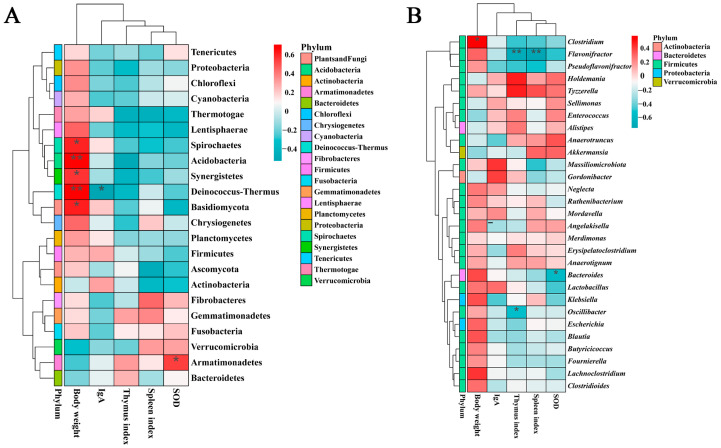
Correlation heatmap analyses between environmental variables and microbial communities. (**A**) Heatmap at phylum level. (**B**) Correlation heatmap of top 29 genera with environmental variables. *: *p* < 0.05, **: *p* < 0.01.

**Figure 6 metabolites-15-00736-f006:**
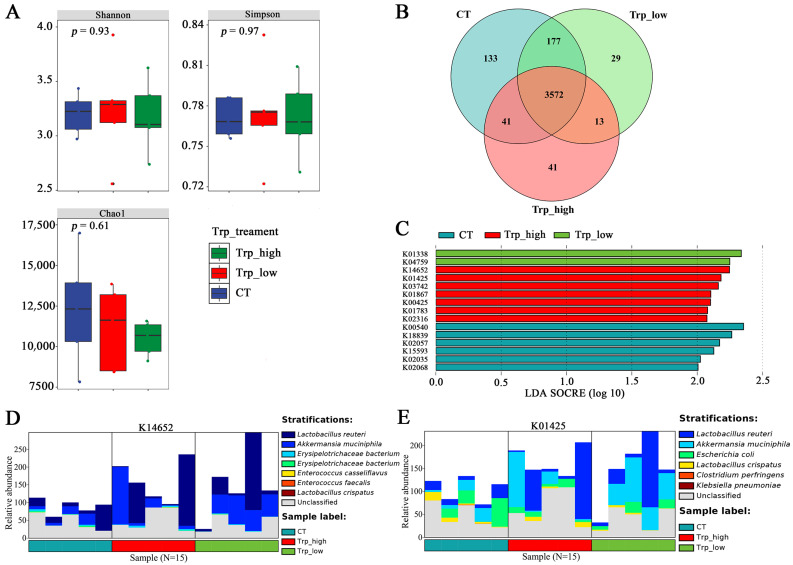
Differential analysis of KEGG orthology between groups of different dietary Trp levels. (**A**) Alpha diversity analyses of KOs under different dietary Trp levels. (**B**) Venn diagram analysis of KOs between groups. (**C**) LEfSe analysis and the KOs with relatively high significance among groups. (**D**,**E**) Barplots of the species origin composition of representative KOs in the Trp_high group; analyses were conducted based on HUMAnN 2.0.

**Table 1 metabolites-15-00736-t001:** Energy and nutrient composition of basal diets (air-dry basis).

Item	Composition, %
Ingredients	
Corn	60.0
Soybean meal	30.0
Fish meal	2.0
Soybean oil	3.0
Premix ^1^	5.0
Total	100
Energy and nutrient composition ^2^	
Metabolizable energy (calculated) ^3^	12.22
Crude protein (calculated)	21.50
Calcium (calculated)	1.00
Available phosphorus (calculated)	0.45
Methionine (measured)	0.46
Lysine (measured)	1.15
Methionine + cysteine (measured)	0.91
Tryptophan (measured)	0.23

^1^ The premix provided the following per kg of diets: VA 10,000 IU, VB1 1.96 mg, VB2 5.76 mg, VB6 3.92 mg, VB12 0.02 mg, VD3 2000 IU, VE 15 IU, VK3 63.8 mg, biotin 0.50 mg, folic acid 0.98 mg, D-pantothenic acid 11.76 mg, nicotinic acid 39.2 mg, Cu (as copper sulfate) 0.80 mg, Fe (as ferrous sulfate) 8 mg, Mn (as manganese sulfate) 10 mg, Zn (as zinc sulfate) 7.52 mg, I (as potassium iodide) 0.04 mg, Se (as sodium selenite) 0.04 mg. ^2^ Amino acids data were from measured values and other nutrient values were derived from calculations. ^3^ MJ/kg.

**Table 2 metabolites-15-00736-t002:** Effect of dietary Trp levels on growth performance of broilers.

Item ^1^	Trp Level, %			*p*-Value
	0.23	0.23 + 0.06	0.23 + 0.12	
BW at day 1 (g)	46.54 ± 0.73	46.79 ± 0.52	46.67 ± 0.72	0.845
BW at day 21 (g)	734.37 ± 23.83 ^a^	719.57 ± 11.31 ^a^	657.87 ± 25.93 ^b^	0.010
BWG (g)	687.84 ± 24.12 ^a^	672.77 ± 10.95 ^a^	611.21 ± 25.97 ^b^	0.010
ADFI (g/d)	44.86 ± 1.37 ^a^	43.29 ± 0.86 ^a^	39.44 ± 0.57 ^b^	0.001
ADG (g/d)	33.40 ± 1.07 ^a^	32.02 ± 0.55 ^a^	29.09 ± 1.25 ^b^	0.005
FCR (g/g)	1.34 ± 0.01	1.35 ± 0.02	1.36 ± 0.07	0.917

^a,b^ Means within the same row sharing the same superscript letter or bearing no superscript indicate no significant difference (*p* > 0.05), and means with different superscript letters indicate a significant difference (*p* < 0.05). Data are presented as mean ± SD (n = 15). ^1^ Abbreviations: BW: body weight; BWG: body weight gain; ADFI: average daily feed intake; ADG: average daily gain; FCR: feed conversion rate.

**Table 3 metabolites-15-00736-t003:** Effect of dietary Trp levels on organ indexes of broilers.

Item	Trp Level, %			*p*-Value
	0.23	0.23 + 0.06	0.23 + 0.12	
Thymus index (g/kg)	3.85 ± 0.42 ^b^	4.43 ± 0.62 ^ab^	4.80 ± 0.66 ^a^	0.011
Liver index (g/kg)	28.04 ± 1.39 ^a^	28.08 ± 3.69 ^a^	24.90 ± 1.66 ^b^	0.026
Bursa of Fabricius index (g/kg)	2.62 ± 0.52 ^ab^	2.97 ± 0.10 ^a^	2.28 ± 0.54 ^b^	0.016
Spleen index (g/kg)	0.76 ± 0.23	0.91 ± 0.17	0.81 ± 0.18	0.303

^a,b^ Means within the same row sharing the same superscript letter or bearing no superscript indicate no significant difference (*p* > 0.05), and means with different superscript letters indicate a significant difference (*p* < 0.05). Data are presented as mean ± SD (n = 15).

**Table 4 metabolites-15-00736-t004:** Effect of dietary Trp levels on serum biochemical indicators of broilers.

Item ^1^	Trp Level, %			*p*-Value
	0.23	0.23 + 0.06	0.23 + 0.12	
GSH-Px (U/mL) ^1^	112.09 ± 3.54	111.58 ± 13.16	105.69 ± 9.02	0.219
SOD (U/mL) ^1^	214.63 ± 20.74 ^b^	290.95 ± 52.00 ^a^	303.96 ± 41.27 ^a^	0.022
T-AOC (U/mL) ^1^	6.82 ± 1.38	6.27 ± 1.04	6.51 ± 1.56	0.718
MDA (mmol/L) ^1^	7.03 ± 0.63	6.59 ± 0.94	6.09 ± 0.78	0.084
IgG (μg/mL) ^1^	17.18 ± 3.19	18.87 ± 4.15	18.79 ± 4.72	0.648
IgA (μg/mL) ^1^	62.82 ± 8.75	65.51 ± 12.90	67.54 ± 9.97	0.680
IgM (ng/mL) ^1^	405.23 ± 80.20	396.17 ± 108.67	396.88 ± 83.39	0.976

^a,b^ Means within the same row sharing the same superscript letter or bearing no superscript indicate no significant difference (*p* > 0.05), and means with different superscript letters indicate a significant difference (*p* < 0.05). Data are presented as mean ± SD (n = 15). ^1^ Abbreviations: GSH-Px: glutathione peroxidase; SOD: superoxide dismutase; T-AOC: total antioxidant capacity; MDA: malondialdehyde; IgG: immunoglobulin G; IgA: immunoglobulin A; IgM: immunoglobulin M.

## Data Availability

The original contributions presented in this study are included in the article/Appendix A. Further inquiries can be directed to the corresponding authors.

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
