# Peer review of "Effect of Tryptophan Supplementation Levels on the Cecal Microbial Composition, Growth Performance, Immune Function and Antioxidant Capacity in Broilers"

_metabolites, 2025, doi:10.3390/metabo15110736_

Round 1

Reviewer 1 Report

Comments and Suggestions for Authors

This study focuses on the effects of tryptophan on the cecal microbiota, growth, and immunity of broilers. It designs a gradient supplementation experiment combined with metagenomic sequencing, and its research direction has practical significance, with core data (e.g., microbiota structure, growth indicators) providing certain support. However, it has issues such as lack of basis for dose selection, undetectable tryptophan metabolites, and unclear recommended doses in conclusions; details of figures/tables and reference formats also need optimization. Overall, details should be improved to enhance scientific rigor and practical guidance.

  1. Rationale for dose selection needs supplementary evidence
    In the Materials and Methods section, low-dose (+0.06%) and high-dose (+0.12%) tryptophan supplementation groups were set, but the basis for determining this dose gradient (e.g., whether it references previous studies on tryptophan nutrition in broilers, pre-experimental results, or industry-recommended standards) was not explained. It is recommended to supplement literature support or pre-experimental data for dose screening to enhance the scientific rigor of the experimental design and avoid subjectivity in dose setting.
  2. Inconsistencies in sample size need clarification
    The Sample Collection section clearly states "15 animals per group (5 replicates, with 3 animals pooled per replicate)", but Table 2 (Growth Performance) is labeled with "n=8", which is a clear inconsistency and may affect data reliability. It is necessary to verify and correct the description of sample size, ensure consistency among "number of replicates - sample size - n-value in statistical analysis", and clarify the correspondence between growth performance and microbial samples in the methods (e.g., whether they are from the same batch of animals).
  3. Functional gene analysis should focus on tryptophan metabolism-related pathways
    KEGG/GO functional analysis mentions that differential KOs/GOs are mainly derived from Akkermansiaand Lactobacillus reuteri, but it does not specifically explain the metabolic pathways corresponding to these differential genes (e.g., tryptophan metabolism-related indole synthesis genes, key enzyme genes in the kynurenine pathway). It is recommended to supplement correlation analysis between differential functional genes and the "tryptophan-microbe-host" interaction (e.g., labeling the expression trends of differential genes in tryptophan metabolism pathways) to avoid disconnection between functional analysis and the research core (effects of tryptophan).
  4. Tryptophan metabolite detection is missing, and mechanism speculation requires caution
    The Discussion speculates that "high tryptophan inhibits feed intake and reduces growth via serotonin", but the concentrations of key tryptophan metabolites (e.g., serotonin, kynurenine, indole derivatives) in serum/intestinal tracts were not detected. This mechanism is only inferred from literature and lacks direct experimental evidence. It is recommended to supplement detection data of such metabolites to verify the mechanism, or clearly state the limitations of this speculation in the Discussion and propose future research directions.
  5. Immune function evaluation indicators need supplementation, and the evidence chain is incomplete
    Table 4 shows no significant differences in humoral immune indicators (IgG, IgA, IgM), with only the thymus index increased, yet the Discussion concludes that "tryptophan enhances immune function"—the evidence is overly insufficient. It is recommended to supplement cellular immune indicators (e.g., peripheral blood T cell subsets, intestinal cytokines) or mucosal immune indicators (e.g., intestinal sIgA content), or discuss "why humoral immunity shows no response while thymus development is affected" to avoid mismatches between conclusions and data.
  6. Experimental cycle is too short, lacking analysis of long-term effects
    Sampling was only conducted at 21 days of age (early stage of broilers). Although "sensitivity during the hyperplasia phase of immune organs" is mentioned, the commercial rearing cycle of broilers is usually 42 days. Results at 21 days of age cannot reflect the long-term effects of tryptophan on late-stage growth performance (e.g., slaughter weight, feed efficiency), intestinal microbial stability, and immune function. It is recommended to supplement sample data at 42 days of age, or clearly state the limitations of early-stage results in the Discussion to enhance the application value of the study.
  7. Reference formats are inconsistent and need standardization
    The reference list has formatting inconsistencies: for example, the separator between authors and titles is incorrect in Entry 43 (Hu et al., 2020), and some entries do not clearly label volume, issue, and page numbers (e.g., Entry 45, Lundberg et al., 2021). It is necessary to standardize all references in strict accordance with the reference format of the Animalsjournal (e.g., Vancouver style) to ensure the completeness and consistency of author information, publication year, title, and journal details.
  8. Recommended dose in the conclusion is unclear, and application value needs strengthening
    The Conclusion proposes "optimizing tryptophan supplementation to balance growth and health", but it does not provide a specific recommended range based on experimental data (e.g., whether supplementing 0.06% on the basis of the basal 0.23% is the optimal dose for "balancing growth and immunity"). It is recommended to clarify the advantage that "low-dose (+0.06%) has no inhibitory effect on growth and increases the thymus index" based on 21-day-old data, and provide specific recommendations for supplementary doses to enhance the guiding significance of the study for broiler farming practices.

Reviewer 2 Report

Comments and Suggestions for Authors
  1. The material and method it very short; it should add more details for clarity. All parameters that you evaluated should be written in detail
  2. Microbiome analysis should be added in more detail and statistical analysis 
  3. The discussion should add more information on the topic: biochemical indicators relating to tryptophan, referring to specific metabolism in broiler and microbiome.    
  4. It should be added a discussion about the correlation of gene expression, metabolic function of microbiome with tryptophan.
Comments on the Quality of English Language

It should be improve

Reviewer 3 Report

Comments and Suggestions for Authors

Comments on the Quality of English Language

Minor correction may be needed.

Round 2

Reviewer 2 Report

Comments and Suggestions for Authors

Accept the revise version fro bublication.

Author Response

Thank you for your comments and we have made further revisions to the manuscript based on the suggestions of the academic editor.